# Extracellular Vesicles Derived from Human Umbilical Cord-Mesenchymal Stem Cells Ameliorate Intervertebral Disc Degeneration

**DOI:** 10.3390/biomedicines13102420

**Published:** 2025-10-03

**Authors:** Sobia Ekram, Faiza Ramzan, Asmat Salim, Marie Christine Durrieu, Irfan Khan

**Affiliations:** 1Dr. Panjwani Center for Molecular Medicine Drug Research, International Center for Chemical and Biological Sciences, University of Karachi, Karachi 75270, Pakistan; sobiaekram25@gmail.com (S.E.); faizakhanmeo@gmail.com (F.R.); asmat.salim@iccs.edu (A.S.); 2University Bordeaux, CNRS, Bordeaux INP, CBMN, UMR 5248, F-33600 Pessac, France; 3Department of Ophthalmology and Visual Sciences, The Aga Khan University, Stadium Road, P.O. Box 3500, Karachi 74800, Pakistan; 4Centre for Regenerative Medicine and Stem Cells Research, The Aga Khan University, Stadium Road, P.O. Box 3500, Karachi 74800, Pakistan; 5Department of Biological and Biomedical Sciences, The Aga Khan University, Stadium Road, P.O. Box 3500, Karachi 74800, Pakistan

**Keywords:** extracellular vesicles, human umbilical cord, mesenchymal stem cells, nucleus pulposus, pain, inflammation, intervertebral disc degeneration, repair

## Abstract

**Background:** Intervertebral disc degeneration (IVDD) is closely linked to low back pain (LBP), a leading cause of disability worldwide. IVDD is characterized by the loss of proteoglycans (PGs), extracellular matrix (ECM) degradation, and reduced hydration of the nucleus pulposus (NP). Extracellular vesicles (EVs) derived from human umbilical cord mesenchymal stem cells (hUC-MSCs) exhibit tissue repair and immunomodulatory effects and are emerging as promising cell-free therapeutics. **Methods:** We established a rat IVDD model via fluoroscopy-guided needle puncture of three consecutive coccygeal discs and confirmed degeneration through Alcian Blue and hematoxylin & eosin (H&E) staining. The gene expression of inflammatory and pain markers (*ADRβ2*, *COMP*, *CXCL1*, *COX2*, *PPTA*, *MMP13*, *YKL40*) was measured by qPCR. Subsequently, we implanted hUC-MSCs or EVs to evaluate their reparative potential. **Results:** Upregulation of inflammatory and pain genes in IVDD was associated with an immunomodulatory response. Tracking DiI-labelled hUC-MSCs and EVs revealed enhanced survival of hUC-MSCs, retention of EVs, and dispersion within rat tail discs; EVs showed greater retention than hUC-MSCs. Implanted EVs were internalized by NP cells and remained within degenerative IVDs. EVs passively diffused, accumulated at the injury site, interacted with host cells, and enhanced function, as shown by increased expression of human chondrocyte-related markers (SOX9, TGFβ1, TGFβ2, COL2) compared to hUC-MSC treatment. Histological analysis of two weeks post-transplantation showed NP cellular patterns resembling chondromas in treated discs. EVs integrated into and distributed within degenerated NP regions, with greater glycosaminoglycan (GAG) content. **Conclusions:** Overall, hUC-MSC EVs demonstrated superior regenerative capacity, supporting a safe, cell-free strategy for disc repair.

## 1. Introduction

Intervertebral disc degeneration (IVDD) is a progressive, inflammation-driven process that causes mechanical and structural failure associated with lower back pain (LBP). LBP caused by degenerative intervertebral discs (IVDs) is a global health concern with a substantial socioeconomic burden [1]. IVDD is a multifactorial condition that disrupts the cellular microenvironment, leading to decreased cellular viability, structural damage, and functional impairment [2]. Although the exact cause of LBP is multifactorial, it is strongly associated with IVDD [3]. The loss of anulus fibrosus (AF) or nucleus pulposus (NP) reduces hydration and depletes glycosaminoglycan (GAG) concentrations and extracellular matrix (ECM) components. The IVDs’ cellular morphology changes due to increased pro-inflammatory mediators, proteoglycans (PGs), and type II collagen degradation inside the NP [4]. Current treatments are largely symptomatic and do not halt disease progression [5]. Conventional treatments, including physiotherapy, analgesics, and anti-inflammatory medications, do not address the disc’s ongoing structural degradation and only provide short-term relief. Although surgical interventions like discectomy and spinal fusion can reduce discomfort, they are intrusive, fraught with risks, and can hasten the deterioration of surrounding segments. Moreover, these methods are unable to restore the disc’s natural biomechanics and cellular milieu [6].

Regenerative medicine represents next-generation therapy techniques. These treatments can be broadly categorized into two approaches: cell-based and cell-free therapies [7]. MSCs may offer longer-lasting advantages because of their persistent secretion and integration. However, the deteriorated intervertebral disc’s extreme microenvironment, acidity, hypoxia, and limited nutrition supply make it harder for cells to survive [8,9]. Furthermore, transplanted cells could put additional stress on the already vulnerable niche. Stem cell-derived extracellular vesicles (EVs) represent a possible substitute for targeted therapy in such settings because, despite their short half-life, they are more stable, less immunogenic, and capable of efficiently delivering regenerative signals without the viability issues connected with living cells [10]

MSCs and their derivatives, especially EVs, have encouraging outcomes in improving the disc microenvironment, lowering inflammation, and helping tissue repair through paracrine signalling [9]. MSCs proliferate and differentiate into specialized cells to repair deteriorated tissue [10]. MSCs can release biologically active molecules in response to their interaction with the host niches, such as platelet-derived growth factor (PDGF), insulin-like growth factor 1 (IGF-1), interleukin 6 (IL-6), matrix metalloproteinase 2 (MMP-2), matrix metalloproteinase 9 (MMP-9), and prostaglandin E2 (PGE2), and thus have anti-fibrosis, anti-apoptotic, anti-inflammatory, angiogenic, and immunomodulatory functions [11,12,13]. MSCs have proven safety and therapeutic implications in the field of cellular therapy [14]. Preclinical investigations have shown MSCs’ efficacy in the treatment of degenerative disc disease [15]. MSCs can differentiate into NP-like cells and aid in the restoration and repair of the normal mechanical function of the disc [16]. MSCs stimulate NP cells to secrete collagen type II, SOX-9, and PGs in the degenerative IVDs [17]. Preclinical studies have demonstrated that MSCs’ paracrine function enhances disc regeneration [18].

EVs have been demonstrated to be effective in immunomodulation and tissue repair [19]. For therapeutic outcomes, this functional persistence is essential, especially in avascular and immunoprivileged locations like the intervertebral disc, where healing requires the continuous presence of regeneration signals [20] MSC-derived EVs have shown promising results due to their nanoscale size [21], low immunogenicity, minimal risk of residual cell survival, long-lasting effects in cell transplantation, and ability to efficiently deliver pro-regenerative and anti-inflammatory signals. Their ease of intercellular transfer enhances their impact on the targeted area [22]. MSC-derived EVs have the potential to stimulate stem cell differentiation [23,24,25]. The use of MSC-derived EVs in IVDD therapy is still in its early stages, but there is significant interest in their application. In contrast to other types of EVs, hUC-MSC-derived EVs have greater therapeutic potential due to their reduced immunogenicity, invasiveness, toxicity, and cost-effectiveness.

The objective of the present work was to assess and compare the regeneration capability of EVs produced from human umbilical cord mesenchymal stem cells (hUC-MSCs) in an in vivo model of IVDD. EVs were separated from serum-deprived hUC-MSC-conditioned medium by the differential centrifugation technique. Either hUC-MSCs or pure EVs were injected intradiscally after IVDD was induced using a rat caudal disc puncture paradigm. Following treatment, the disc tissues were evaluated using molecular, histological, and immunohistochemical investigations to determine the expression of ECM components and markers specific to notochordal cells (NCs). The restoration of disc structural and functional integrity was also assessed by biomechanical testing, which enabled us to investigate the possibility of using hUC-MSC-derived EVs as a practical, cell-free substitute for IVD regeneration

## 2. Materials and Methods

### 2.1. Ethical Statement

Adult male Wistar rats, weighing 200–300 g and 2–6 months of age, were housed at the Animal Resource Facility of Dr. Panjwani Center for Molecular Medicine and Drug Research, University of Karachi. The rats were placed in labelled cages and given food and sterile water ad libitum in a facility that was kept at 21 ± 1 °C with a relative humidity level of 55 ± 5%, and 12–12 h standard light and dark cycles. The animal procedures were carried out in compliance with international standards for laboratory animal care and use, along with the animal study protocol, and approval was obtained from the Institutional Animal Care and Use Committee (IACUC) of ICCBS. The protocol number for the study was 2017-0051.

### 2.2. hUC-MSC and hUC-MSC Derived EV Isolation

The study obtained informed consent from human subjects for the procurement of human umbilical cord (hUC) tissues. MSCs were isolated from the hUC tissue using established protocols [26]. The detailed methodology of isolating MSCs from hUC tissue and their characterization was performed according to the previously reported [26,27]. EVs were subsequently isolated from the conditioned medium of hUC-MSCs employing a sequential centrifugation method [28]. All EVs used in this study were derived from a single donor hUC-MSCs line, with three independent isolations prepared from the same cell culture to ensure consistency across experiments. In brief, hUC-MSCs were cultured in complete DMEM until reaching 80% confluence. The culture medium was replaced with serum-free DMEM, and cells were incubated for 48 h at 37 °C in 5% CO_2_. Following the incubation period, the conditioned media rich in EVs were collected and subjected to centrifugation at 500× *g* for 10 min, followed by 800× *g* for 10 min to eliminate cellular debris. The resultant supernatant was further centrifuged at 2000× *g* for 20 min, followed by an additional 2000× *g* centrifugation for 30 min to remove residual cellular components. Throughout these steps, all centrifugations were conducted at 4 °C to preserve EV integrity. Cell-free supernatants were subsequently transferred to fresh Falcon tubes and centrifuged at 15,000× *g* for 30 min at 4 °C to pellet the EVs. The harvested EV pellet was washed with sterile PBS.

### 2.3. hUC-MSC Derived EV Characterization

The EVs were subjected to comprehensive characterization based on their size and morphology. This assessment employed a range of analytical techniques, including scanning electron microscopy (SEM), dynamic light scattering (DLS), atomic force microscopy (AFM), and fluorescence microscopy.

### 2.4. Scanning Electron Microscopy (SEM)

As EVs exhibit nanoscale dimensions, their detection necessitates the utilization of high-resolution microscopy techniques. SEM emerges as the principal modality for EV visualization. A 10 µL aliquot of the EV suspension underwent fixation with 4% paraformaldehyde (PFA) for 10 min. Subsequently, the fixed EVs were diluted in distilled water (1:5), and 5 µL of the resulting solution was applied onto a coverslip and air-dried at room temperature. To mitigate charging artefacts and enhance the secondary electron signal, a thin layer of 15 nm conductive gold was deposited onto the EV sample utilizing an auto-fine coating chamber (JEC-3000FC, Jeol, Tokyo, Japan) programmed at 20 mA current for 30 s. Observation of EV images was conducted using a Jeol scanning electron microscope (JSM-IT 100 SEM Jeol, Tokyo, Japan) operating at an accelerating voltage of 10 kV under vacuum conditions. Image analysis of EVs was executed utilizing dedicated software (SEM Control user interface version 7.11C).

### 2.5. Dynamic Light Scattering (DLS)

The size distribution and zeta potential of EVs derived from hUC-MSCs were assessed using the Zetasizer DLS Nano ZS system version 7.12 (MAL1160489, Malvern Instruments, Malvern, UK) according to the manufacturer’s guidelines. For the analysis of EV pellets, they were dissolved in 1 mL of sterile phosphate-buffered saline (PBS). The DLS instrument settings included 7 runs of 20 s each to ensure precise determination of the hydrodynamic size distribution and stability of the EVs.

### 2.6. Atomic Force Microscopy (AFM)

The size distribution of EVs was further examined utilizing an AFM, the model AFM-5500 (Agilent Technologies, Santa Clara, CA, USA). For the analysis, EV pellets were diluted in sterile 1X PBS at a ratio of 1:10. Subsequently, 10 µL of the resulting solution was applied onto a freshly cleaved mica sheet and incubated overnight at 37 °C to promote evaporation. Scanning measurements were conducted in the air using silicon nitride probes by AFM manufactured by Agilent Technologies, operating in amplitude modulation (AC) or tapping imaging mode. Image processing and analysis were executed utilizing Pico View 1.12.2 software to elucidate the size distribution and morphology of the EVs in detail.

### 2.7. HUC-MSC Derived EV Labelled with DiI via Fluorescence Microscopy

EVs derived from hUC-MSCs were labelled using the 1,1’-dioctadecyl-3,3,3′,3′-tetramethyl indocarbocyanine perchlorate (DiI) lipophilic cationic membrane-binding fluorescence dye (V-22885, Vybrant^®^ DiI cell-labelling solution, Invitrogen, Carlsbad, CA, USA). Initially, the EV pellet was resuspended in 5 μL of DiI-labelling dye (10 μM) provided in the kit. This solution was immediately mixed with 1 mL of serum-free medium and incubated for precisely 7 min at 37 °C in a 5% CO_2_ incubator to ensure uniform labelling. The labelling process was terminated by adding 5 mL of complete media. Following labelling, the EV suspension underwent two washes with PBS (1X) and was subsequently centrifuged at 16,000× *g* for 30 min at 4 °C. The resulting EV pellet was resuspended in 100 μL of 1X PBS. Visualization of DiI-labelled EVs was performed using a fluorescence microscope (Nikon Ti-2, Tokyo, Japan), and image analysis was conducted using NIS-Elements AR version 5.01 software.

### 2.8. Experimental Groups

IVDD was induced in non-contiguous coccygeal IVDs of the rat tail. A total of 60 adult Wistar rats were used in vivo study and randomly divided into four experimental groups: Normal disc as a control (*n* = 9), IVDD control (*n* = 21), hUMSC-treated (*n* = 15), and hUC-MSCs derived EV-treated (*n* = 15). Each animal contributed two coccygeal IVDs (Co5/6, and Co6/7), allowing for sufficient tissue allocation across different experimental analyses. The Co5/6 and Co6/7 positions were selected for their accessibility, anatomical consistency, and established use in reproducible needle puncture IVDD models in rats.

For histological and immunohistochemical analysis, IVDs from nine rats (3 discs per group) were harvested and processed accordingly. For biochemical assays, including water content and glycosaminoglycan (GAG) quantification, IVDs from another eighteen rats (3 discs per group) were used, that divided equally between the two assays. Importantly, each animal was assigned to a specific endpoint to avoid overlapping data sources across different analyses. For gene expression analysis, animals in each experimental group were euthanized at different time intervals following needle puncture injury (21-G syringe): on Days 1 (*n* = 3), 2 (*n* = 3), 5 (*n* = 3), and 10 (*n* = 3). Transplantation of hUC-MSCs and hUC-MSC derived EVs was performed on Days 2 and 5 (*n* = 3 each). Discs Co5/6 (*n* = 3) from the rat coccygeal tails were preserved intact to serve as normal, uninjured controls. This distribution strategy allowed for time-course analysis of gene expression with three discs analyzed per time point per group, and each sample was analyzed in technical triplicate to ensure maintaining consistency and integrity across all the assay groups.

### 2.9. Development of an IVDD Model

Surgical intervention was performed following established procedures [29,30]. Briefly, rats were anesthetized using an optimum dose of 60 mg/kg of ketamine hydrochloride and 7 mg/kg of xylazine through intraperitoneal (IP) injection. The “Tail Pinch Assessment” was performed to ensure optimal unconsciousness by checking the animal’s reflexes. During all experiments, two successive coccygeal IVDs of the tail at positions Co-5/6 and Co-6/7 were specified. The Co-5/6 IVDs were used as non-punctured controls. To induce disc degeneration, the skin of the rat tail was disinfected with 70% ethanol, and rats were placed in a prone position. Under fluoroscopic guidance, a Co-6/7 IVD was punctured using a 21-gauge needle. Through the middle of the NP, the spine was punctured from ventral to dorsal. The needle’s penetrating depth was regulated Via a clamp set at 5 mm to penetrate the core of the NP and then twisted 180 degrees and held for 10 s. After causing degeneration, the needle was removed, and the puncture disc position was properly labelled. Before the animals recovered from anesthesia, they were returned to their appropriate cages. The rats were regularly monitored and followed the standard postoperative treatments.

### 2.10. hUC-MSC and MSC-Derived EV Labelling for Transplantation

In the rat model of IVDD, hUC-MSCs were administered at a concentration of 1 × 10^6^ cells/100 µL, and the vehicle used was sterile PBS. For hUC-MSC-derived EVs, they were injected at an equivalent concentration of 100 µg of total protein content per 100 µL, based on BCA protein quantification. The EVs were also suspended in sterile PBS. The selected dosages were based on previously published studies demonstrating therapeutic efficacy in IVDD models and optimized through preliminary dose-finding experiments. Cell culture flasks containing approximately 1 × 10^6^ hUC-MSCs and 5 × 10^6^ hUC-MSC derived EVs, respectively, were used therapeutically for both the hUC-MSC and EV treated groups. MSCs were rinsed with PBS, trypsinized with 0.25% trypsin, and hUC-MSCs and EVs pellets after centrifugation, as discussed in the isolation section that were labelled with DiI membrane labelling fluorescent dye (V-22885, Vybrant^®^ DiI cell-labelling solution, Invitrogen, USA) for 20 min following the standard procedure. Finally, labelled hUC-MSCs and EV pellets (*n* = 3 each) were washed and properly resuspended in 100 µL of sterile PBS for in vivo transplant. MSCs and EVs were fluorescently tagged with red DiI dye, which possessed excitation and emission at 553/570 nm wavelengths, to track them in vivo after transplantation in the IVDD rat models.

### 2.11. MSC and EV Transplantation into an IVDD Model

In this study, the transplantation (*n* = 6) was performed immediately after the IVDD induction, to simulate an early intervention strategy, aiming to modulate the acute inflammatory response, enhance hUC-MSCs and EV integration, and prevent progressive degeneration, in line with established rat tail IVDD models (*n* = 3) for 2 weeks to better mimic the pathological conditions of established IVDD [29,30]. For transplanting hUC-MSCs and hUC-MSC derived EVs, a sterile 100-unit syringe (30 G × 5/16”; 0.3 mm × 8 mm) was used. Following the induction of IVDD in the animal, a 100 µL suspension of labelled hUC-MSCs or EVs was immediately injected into the rat’s intradiscal coccygeal Co-6/7 spine. Unlabeled hUC-MSCs and hUC-MSC derived EVs were also transplanted into other rat IVDD models. Rats were observed daily for 2 weeks after transplantation.

### 2.12. Histological Analysis

After two weeks of surgery, all animals were euthanized using an overdose of sodium phenobarbital (≥150 mg/kg, intraperitoneally) by international standards for laboratory animal care and use and approved ASP. The Co-5/6 and Co-6/7 IVDs of the experimental animals were precisely harvested. The intact disc was cleaned, and any remaining bones were carefully removed from both edges of the IVDs. The residual calcified tissue remained tightly associated with the disc region, particularly around the vertebral endplates. Each disc segment was demineralized for 1 h in 11% formic acid for histological analysis to balance effective softening with preservation of antigen integrity. The demineralized IVDs segment was then placed directly into a mold, stood upright with the disc’s base facing upward, and was completely coated with optimal cutting temperature (OCT) medium (Surgipath, FSC22, Leica Microsystems, Bannockburn, IL, USA), and placed at room temperature for 2 h to eliminate any bubbles. Lastly, the molds were instantly frozen at −20 °C. Cryosectioning of frozen tissue block was achieved by using a cryotome E (Shandon, Thermo Electron Corporation, Altrincham, UK) machine. The frozen tissue block was removed from the mold and placed on an OCT-coated cassette. A newly sharpened microtome-cutting blade was used to cut 8 µm thick sections. Sections were then transferred to gelatin-coated microscopic slides (2105, SuperFrost, Roth, Germany), followed by staining with Hematoxylin and Eosin (H&E) and Alcian Blue staining as per the manufacturer’s procedure. The images were acquired using a bright field microscope (NiE, Nikon, Japan) and analyzed by NIS Element-AR software.

### 2.13. Immunohistochemistry

Tissue samples were collected from rat coccygeal spine IVDs after 2 weeks of post-treatment to examine the early cellular and molecular responses following IVDD and therapeutic intervention. The two weeks short-term timeline allowed us to observe hUC-MSCs or EV localization and modulation of early degenerative changes. Immunohistochemical analyses were performed on *n* = 3 tissue sections per animal. A total of *n* = 9 independent experiments were conducted. Immunohistochemical analysis of cryosectioned IVD explant tissue was conducted to assess the functional activity of transplanted, labelled hUC-MSCs and EVs using human antibodies against SOX-9, TGFβ-1, TGFβ-2, and COL-2. The cryosection slides were fixed by incubating with 4% paraformaldehyde (PFA) for 10 min, followed by permeabilization, blocking, and incubated with specific primary antibodies against SOX-9 (Polyclonal Antibody, PA5-81966, Invitrogen, USA), TGFβ-1 (Polyclonal Antibody, Y058105, Applied Biological Materials Inc., Richmond, BC, Canada), TGFβ-2 (Monoclonal antibody, Y300232, Applied Biological Materials Inc., Canada), and COL-2 (Monoclonal antibody, MA5-12789, Invitrogen, USA) targeting human nucleus pulposus cells at a dilution of 1:50. The staining was observed using goat anti-rabbit Alexa Fluor 488 secondary antibody (Polyclonal antibody, AB_2338051, Jackson ImmunoResearch Inc., West Grove, PA, USA) for SOX-9, TGFβ-1, TGFβ-2, and anti-mouse Alexa Fluor 488 secondary antibody (Polyclonal antibody, AB_2338858, Jackson ImmunoResearch Inc., USA) for COL-2 in a 1:100 dilution for 1 h at 37 °C. To stain the nuclei, slides were counterstained with 5 ng/mL 4′, 6-diamidino-2-phenylindole (DAPI) in PBS for 10 min at room temperature. The images were captured under a fluorescence microscope (ECLIPSE Ni-E, Nikon, Japan) and then processed with Adobe Photoshop software version 26.10. The fluorescent intensity of the specified marker and its expression were quantified using Image J software version 1.54p by two independent, blinded investigators. For each sample, three mid-sagittal sections were evaluated, and three consistent high fluorescent intensity per field within the IVDs (NP and AF regions) were selected. Quantitative data were normalized to the region of interest and compared against the hUC-MSCs group as a reference.

### 2.14. Radiographic Analysis

The disc height index (DHI) was determined Via radiographic examination [30]. Briefly, a sustained degree of muscular relaxation was maintained in each animal at each temporal point during radiographic analysis, and intensive care was taken to retain a persistent level of anesthesia. The animals were placed in a prone posture on a plastic slot plate with their tails upright. The pre-treatment radiographs were recorded as a baseline assessment. Under anesthesia, X-ray images of IVD levels such as Co-5/6 and Co-6/7 in all experimental groups were captured by applying a 66 cm radiopaque scale, 30 kV of penetration power, and 40 mA in 40 sec of exposure time through the Glenbrook Technologies Bench-Top LabScope instrument (GTI-2000, Glenbrook Technologies, Randolph, NJ, USA). Radiographs were acquired prior to and during the following 2 weeks of index surgery. Radiographs were taken in triplicate and digitally recorded with fluoroscopic image processing software. The DHI was computed by averaging the measures acquired from each coccygeal position of the adjacent cartilaginous end point width and dividing them by three (DHI = (DH1 + DH2 + DH3)/3) of contiguous coccygeal disc height in triplicate. Changes in the DHI of the experimental groups were represented as a percentage of the normal control IVD’s height (DHI percentage = (DHI experimental group/DHI control) multiplied by 100) under the previously established procedure [30].

### 2.15. Nucleus Pulposus Water Content Analysis

NP wet weight changes were measured in milligrams in triplicate and reported as percentages. The post-operative IVDs (*n* = 12) were harvested after two weeks of surgery from the rat tails, and disc tissue was completely excised and kept in pre-weighed vials. The IVD tissues were then placed in vials after being weighed and incubated overnight in the oven. The next day, the changes in NP wet weight were measured through an analytical balance with ±0.03 mg in triplicate. The ratio of NP water content before and after IVD tissue placement in the oven was calculated in milligrams. Percentages were calculated by the formula:H_2_O content in % = (wet weight − dry weight)/wet weight × 100.(1)

### 2.16. Glycosaminoglycan (GAG) Content Analysis by Chondroitin Sulphate

GAGs were isolated from dried IVD tissues of a rat tail by digesting them for 6 h at 70 °C with pepsin (CAS: 9001-75-6, Sigma, St. Louis, MO, USA). After digestion, tissue lysates were vortexed several times before being centrifuged. Then, the lysate in a 96-well plate was incubated with 1,9-dimethylmethylene blue (DMMB) dye (Sigma) in a glycine/NaCl solution at pH 3.1, and thus the complex produced was measured spectrophotometrically at 525 nm wavelength. Total GAG content analysis was calculated from standard curves using chondroitin-6-sulphate extracted from shark cartilage (C4384-250MG, Sigma-Aldrich, USA) as a reference and normalized to sample dry weights. The analysis was carried out in triplicate.

### 2.17. mRNA Extraction and Transcriptional Profiling

Gene expression analysis of pain and inflammatory markers was performed on rat IVDs when acute inflammation was induced. The animals in each experimental group were euthanized with an overdose of sodium phenobarbital at different time intervals, namely Days one (*n* = 3), two (*n* = 3), five (*n* = 3), and ten (*n* = 3) days after needle puncture injury with a 21-G syringe, and transplanted hUC-MSCs and hUC-MSC derived EVs at days two (*n* = 3) and five (*n*= 3). The rat-coccygeal tail’s discs Co5/6 (*n* = 3) remain intact as normal discs. The entire IVDs of specified coccygeal segments Co5/6 and Co6/7, along with contiguous vertebrae or spines, were excised, and the collected IVDs were homogenized Via a tissue homogenizer (Ultra-Turrax, IKA-Werke, Staufen, Germany). At different time points, RNA was extracted from disc tissue of normal, punctured, and transplanted hUC-MSCs and hUC-MSC derived EVs for gene expression analysis. The markers *ADRβ2*, *COMP*, *CXCL-1*, *COX-2*, *PPT-A*, *MMP-13*, *YKL-40*, and *β-actin* were used in this study, were synthesized by Integrated DNA Technologies (IDT), USA. Their details are mentioned in Table 1. The expression of pain and inflammatory primers was normalized to that of *beta-actin*. RNA was isolated from IVD tissue samples using the Trizol (15596026, Invitrogen, Carlsbad, CA, USA) method, and RNA concentration and purity were assessed using a NanoDrop spectrophotometer (ND-2000, Thermo Fisher Scientific, Waltham, MA, USA). To synthesize cDNA, the RevertAid^TM^ First Strand cDNA Synthesis Kit (K1622, ThermoScientific, USA) was used, which was then amplified using qPCR master mix (A600A, Promega, Madison, WI, USA) on a CFX96 Touch Real-Time PCR detection system (1854096, Bio-Rad Laboratories, Hercules, CA, USA).

### 2.18. Statistical Analysis

The data are represented as the mean ± standard error mean (SEM) or standard deviation (SD) of biological triplicates. ANOVA and Bonferroni post hoc tests were used to determine statistical significance for comparisons with 95% confidence intervals (*p* ≤ 0.05) among multiple pairwise groups through SPSS Statistics software (IBM, version 21.0). An independent *t*-test was used to determine statistical significance differences in means for comparisons with *p* ≤ 0.05 between the two groups through Microsoft Excel. The graphs were statistically evaluated in GraphPad Prism 5 software.

## 3. Results

### 3.1. Characterization of hUC-MSC and EVs

MSCs were characterized as we previously reported [31,32,33,34]. The characterization of EVs isolated through the differential centrifugation technique was conducted using a range of microscopic and analytical techniques. SEM imaging at a magnification of 5000× allowed for the examination of EV surface morphology, revealing rounded structures with diameters ranging from 0.300 to 0.800 μm, thereby contributing to the understanding of EV physical attributes, as shown in Figure 1A. High-resolution imaging Via AFM at a magnification of 1 µm was employed to discern EV topography and size distribution, offering further insights into their physical properties (Figure 1B). DLS analysis provided information on EV size distribution in solution, yielding a reported hydrodynamic diameter of 403.3 ± 85.94 nm and a *z*-potential of −0.270 ± 4.02 mV, indicative of EV surface charge (Figure 1C). Fluorescent labelling DiI dye, a lipophilic membrane fluorophore, EVs were labelled for visualization under fluorescence microscopy, facilitating the analysis of their morphology Figure 1D. The comprehensive utilization of these methodologies facilitated a detailed characterization of the isolated EVs, encompassing morphological features, size distribution, surface properties, and molecular composition.

### 3.2. Radiographic Analysis of the IVDD Model

IVDs were punctured with a 21-gauge needle into the healthy coccygeal 6/7 disc of the rat tail under a fluoroscope. Figure 2A depicts a representative needle route for IVD puncture Via the needle puncture technique. The use of radiographic image records aided in the appropriate positioning of the needle into the spinal discs and measured the mean DHI of the punctured disc. Radiographs of the coccygeal disc were acquired with the animals euthanized 2 weeks after transplanting, and their disc height was measured before sacrifice and spine harvest for IVD analysis, as shown in Figure 2B.

### 3.3. qPCR Analysis of Pain and Inflammatory Genes After the Degeneration of IVD

To validate the IVDD rat model, the total comparative study of pain and inflammatory gene expression profiles at 1, 2, and 5 days was examined by qPCR in both control and IVD-degenerated groups. The transcriptional analysis revealed that the IVDD model was successfully developed. On day 1, inflammatory genes *COMP*, *CXCL-1*, *COX-2*, *PPT-A*, and *MMP-13* showed significant upregulation, while on day 2, *ADRβ2*, *COMP*, *CXCL-1*, *COX-2*, *PPT-A*, *MMP-13*, and *YKL-40* genes displayed higher expression in damaged coccygeal IVDs of a rat degenerated model when compared to a normal control, as well as inflammatory genes *ADRβ2*, *CXCL-1*, *COX-2*, *PPT-A*, and *YKL-40* increased significantly on day 5 (Figure 2C). The expression of *COMP* and *COX-2* was significantly increased on day 1 with *p* < 0.01 (**) and *p* < 0.001 (***), respectively, while *CXCL-1*, *PPT-A*, and *MMP-13* were significantly greater on day 2 with *p* < 0.001 (***). Inflammatory markers, including *ADRβ2*, and *YKL-40*, were significantly upregulated on day 5 with *p* < 0.001 (***) when compared to those in normal discs. Each inflammatory gene bar graph compares the levels of expression of normal IVDs and IVDD models, as illustrated in Figure 2C. These findings confirmed the induction of a degenerative and inflammatory microenvironment following disc injury.

### 3.4. Histological Analysis

At 2 weeks after transplanting, H and E staining revealed that the IVDD group exhibited a highly disrupted NP structure and a serpentine arrangement of AF. With fewer chondrocytes, the boundary between NP and AF was entirely ruptured, as was fissure formation in the NP region. In contrast, the hUC-MSCs and EVs groups displayed a relatively well-restored NP, a distinctive NP/AF boundary, and chondrocyte-like cells in the NP area. The EVs group exhibited a chondrocyte-like cellular arrangement, which differed from punctured discs and enhanced ECM proteoglycan deposition [35]. Furthermore, transplanted EVs in the NP area demonstrated that cellularity and NP morphology improved more in the discs that received EVs. Alcian Blue staining in the hUC-MSCs and EVs groups revealed deposition of ECM-rich proteoglycan, as shown in Figure 3. The implanted EVs were integrated and distributed in the damaged regions of the NP and had a greater glycoprotein content than the deteriorated discs. In the EVs treatment group, IVD degeneration was significantly lower than in the control group. These findings implied that EVs transplanted into discs had a better ability to restore the NP (Figure 3).

### 3.5. In Vivo Tracking of Transplanted hUC-MSCs and EVs Through Fluorescence Microscopy

The survival of transplanted hUC-MSCs and EVs was tracked by DiI-labelled dye and evaluated using a fluorescence microscope at a magnification of 10× after two weeks of transplantation. The fluorescence intensity increased significantly (** = *p* < 0.01) when EVs were compared to the hUC-MSC group. The labelled cells and vesicles demonstrated that hUC-MSCs and hUC-MSC derived EVs had higher survival, diffusion, and dispersion in rat degenerated discs; however, EVs had significantly greater accumulation than normal hUC-MSCs (Figure 4A,B).

### 3.6. Immunohistochemistry

Protein expression analysis was carried out to analyze the co-localized expression of human chondrogenic markers targeting SOX-9, TGFβ-1, TGFβ-2, and COL-2 with lipophilic cationic fluorescent labelling of indocarbocyanine DiI dye. As shown in Figure 5A,B, immunohistochemical examination revealed that these EVs aid in the differentiation of endogenous MSCs into the NP of the IVDs. According to the fluorescence intensities per field, EVs had significantly higher expression of chondrocyte proteins, as shown in Figure 5C. These transplanted EVs survived for a long period, diffused into the deteriorated IVDs more than normal hUC-MSCs, and expressed greater levels of human SOX-9, TGFβ-1, TGFβ-2, and COL-2 than normal hUC-MSC-implanted discs. This finding suggests that the transplanted EVs distributed passively Via diffusion through the ECM survived and showed aid in differentiation towards functional NP lineages.

### 3.7. Disc Height Index Assessment

Radiographs of the caudal disc were captured under anesthesia. Two weeks after transplantation, the mean % DHI showed that the DHI was significantly preserved in the EV-treated group, surpassing the hUC-MSC group and far exceeding the degenerated group. The differences in % DHI were significant during the 2 weeks of post-transplantation Figure 6A.

### 3.8. Water Content Analysis

A heat evaporation technique was used to determine the water content of in vivo groups. The physicochemical characteristics of the water content in the NP of harvested discs were examined to evaluate the effects of hUC-MSC and EV transplantation. The water content in the IVDD group was much lower than in the control IVD group. The water content of punctured discs was lowered by 47% by weight in mg when compared to the normal water content of 55%. The NP of the needle punctured IVDs was slightly reduced compared to the normal IVDs, which might be attributed to the single event of needle puncture. The EVs groups displayed significantly greater quantified water content than the IVDD and hUC-MSCs groups. The water content in NP increased significantly by 11% and 15% in punctured IVDs receiving hUC-MSCs and EVs, respectively (Figure 6B). Injected EVs had a more significant effect than hUC-MSCs, restoring the nucleus pulposus water content to near or above normal.

### 3.9. Glycosaminoglycan (GAG) Content Analysis

GAG analysis Via DMMB assay enabled the detection of chondroitin sulphate, the major GAG per milligram expressed in IVDs. A significantly higher level of GAGs was detected in the normal disc and hUC-MSC and EV treated discs (* = *p* < 0.05 and ** = *p* < 0.01, respectively) when compared to IVDD (*** = *p* < 0.001), as shown in Figure 6C. The DMMB findings supported the evidence of the EVs showing stronger GAG recovery, reflecting enhanced matrix regeneration.

### 3.10. Transcriptional Profiling

We further assessed how hUC-MSCs and EVs influence inflammation at the molecular level. The transcription of rat-specific pain and inflammatory markers involving *ADRβ2*, *COMP*, *CXCL-1*, *COX-2*, *PPT-A*, *MMP-13*, and *YKL-40* in transplanted hUC-MSCs and EVs was examined using qPCR after 2 and 5 days, as shown in Figure 7. The findings indicated that both hUC-MSC and hUC-MSC derived EV transplanted groups showed pain and inflammatory markers downregulation, while EVs expressed most of these markers at reduced levels compared to hUC-MSCs. Gene expression analysis revealed that EVs significantly downregulated the expression of *ADRβ-2* and *CXCL-1* (day 2; ** = *p* < 0.01 and day 5; * = *p* < 0.05) when EVs were compared with normal hUC-MSCs. Furthermore, *COX-2* and *PPT-A* expressions were significantly lower (*** = *p* < 0.001) on days 2 and 5 following EV regeneration. *COMP* gene expression was significantly higher in both transplanted groups (MSCs at days 2 and 5 had * = *p* < 0.05 and ** = *p* < 0.01, respectively, while EVs at days 2 and 5 had ** = *p* < 0.01 and *** = *p* < 0.001, respectively). Interestingly, COMP, a structural matrix gene, was upregulated in both groups, with EVs inducing the strongest expression, suggesting active matrix remodelling. Furthermore, *MMP-13* expression was reduced (* = *p* < 0.05) in EVs derived from hUC-MSCs at the end of the day 2 regeneration phase, except for EVs on day 5, where no significant difference was observed. *COMP* and *MMP-13* did not reveal any significant expression on day 2 in the IVD degeneration group. Furthermore, *YKL-40* expression was significantly lower in EVs on days 2 and 5 (*** = *p* < 0.001) than in hUC-MSCs. These findings suggest that hUC-MSCs and hUC-MSC derived EVs may regulate the immune reaction and lower the levels of expression of inflammatory mediators in the acute inflammatory process of degenerative IVDs. These findings indicate that EVs not only reduce inflammatory gene expression more effectively but also support early disc regeneration by promoting anabolic pathways and structural integrity.

Together, these molecular, histological, and functional results demonstrate that hUC-MSC-derived EVs provide superior therapeutic benefit compared to hUC-MSCs alone in restoring disc structure, reducing inflammation, and promoting early regenerative signalling in a rat model of IVDD.

## 4. Discussion

IVDD is a complicated, multifactorial degenerative condition influenced by environmental, physical, and hereditary factors, affecting millions of individuals globally. MSCs have been classified as a promising therapeutic option for IVDD-related LBP, they enhance cellular proliferation [36], reduce oxidative stress [37], and suppress apoptosis [38]. EVs derived from MSCs play a critical role in the treatment of NP abnormalities, particularly in the early stages of degeneration. In the field of regenerative medicine, EVs are effective tools for biomolecule delivery and intercellular communication. When compared to cell therapies, EVs are more beneficial as they have a lower risk of tumorigenicity, immune reactions, and toxic outcomes [39]. Furthermore, due to their smaller size and reduced complexity, EVs are easier to produce, store, and preserve. EVs hold significant potential to replace MSCs in treating musculoskeletal degenerative conditions such as IVDD. They are typically internalized by recipient cells, where they deliver bioactive molecules that induce phenotypic changes. However, rigorous preclinical studies using animal models that closely resemble human IVD ageing and degeneration are required. The present study utilizes a rat model to characterize IVD degeneration, providing a platform to explore the mechanisms underlying degeneration and to evaluate the safety and efficacy of EVs as a cell-free therapeutic strategy.

The isolated hUC-MSCs showed typical spindle-shaped fibroblast-like morphological features and expressed MSC-specific markers, including CD29, CD44, CD73, CD90, CD105, CD117, STRO-1, Vimentin, and Lin-28. In contrast, they were negative for hematopoietic markers such as CD45 and HLA-DR. In vitro trilineage differentiation confirmed their potential to differentiate into chondrocytes, osteocytes, and adipocytes. We have reported these findings previously [26,27,28,29,31]. EVs obtained from hUC-derived multipotent MSCs enhance pro-angiogenic [35,40], anti-apoptotic [41,42], anti-fibrotic activity [43], and immunomodulatory properties [44]. After isolation and characterization of hUC-MSCs, the cells were cultured to obtain EVs from conditioned or serum-free media. EVs were isolated using differential centrifugation, and multiple methods were employed to characterize them. Scanning electron microscopy (SEM) at various magnifications revealed spherical particles ranging from 300 to 1000 nm. Atomic force microscopy (AFM) provided topographic imaging, confirming the size distribution. Dynamic light scattering (DLS) measured the average size as 403.3 ± 85.94 nm and the zeta potential as −0.270 ± 4.02 mV. Purified EVs were tagged with DiI membrane dye and examined using fluorescence microscopy. Overlaying phase-contrast and fluorescence images confirmed precise colocalization, indicating that the observed signal was derived from labelled EVs and not from dye aggregates. Western blot analysis confirming the presence of EV-specific markers CD81 and Annexin V was previously reported in our relevant study [28]. These findings are consistent with the MISEV2018 guidelines, supporting the classification of these isolates as EVs.

EVs support disc regeneration by regulating inflammation and the restoration of the ECM. They transport an intricate array of bioactive molecules, such as growth hormones, cytokines, and anti-inflammatory microRNAs (e.g., miR-21, miR-146a), which can inhibit pro-inflammatory signalling pathways like NF-κB and lower the production of catabolic enzymes like MMPs and ADAMTS. Native disc cells are preserved, and matrix degradation is decreased by EVs, which change the local environment from a pro-inflammatory to a reparative state [6,45]. Furthermore, the expression of ECM proteins, including collagen type II and aggrecan, necessary for preserving disc hydration and structure, EVs promote anabolic processes. Because of their dual function of immunomodulation and tissue regeneration, EVs are positioned as powerful biological modulators that can treat the underlying pathology and symptoms of IVDD. Additionally, because they are nanosized, resident disc cells can better absorb them into the tissue, guaranteeing the effective and focused distribution of regenerative signals [6].

In this study, histological investigation revealed increased ECM content and distribution, as well as the existence of glycoproteins and proteoglycans, suggesting that hUC-MSCs and hUC-MSC derived EVs altered the metabolic activity of rat discs. H and E staining of hUC-MSCs and EVs, two weeks after transplantation, exhibited cellular patterns like chondromas in the NP area, which differ from degenerated discs. Furthermore, transplanted hUC-MSCs and EVs in the NP area showed that the cellular framework and NP composition advanced more rapidly in the IVDs that received EVs. Alcian blue staining revealed that the implanted EVs integrated and distributed within damaged regions of the NP, which had greater GAG protein levels than the degenerative discs. When compared to the control group, hUC-MSC and EV treatment significantly reduced IVD degeneration. These findings suggested that EVs transplanted into discs had a better ability to repair the NP. These outcomes agree with the previous findings that showed improvements in cellularity and GAGs using cell-free therapy in a variety of animal models [38,46,47]. Similarly to our findings on EVs, few reports have shown improvement in cellular concentration and matrix synthesis Via transplanting human BM-MSCs [48], urine-derived stem cells [49], placenta-derived mesenchymal stem cells [47], and induced pluripotent stem cell (iPSC)-derived MSCs [50], based mainly on histological evidence, but no in vivo studies have shown improvement in the quantitative evaluation of GAG concentration following human cell transplantation.

The co-localized expression of chondrogenic-specific human antibodies (SOX-9, TGFβ-1, TGFβ-2, COL-2) with cell membrane labelling DiI dye was significant in EV recipient IVDs. In hUC-derived MSC treatment, a minimal rise in human chondrocyte-specific markers was observed. This suggests that signalling induction from the disc’s microenvironment was crucial in modulating the expression of chondrocyte-specific proteins in EVs rather than hUC-MSCs. In comparison to normal hUC-derived MSCs, these transplanted EVs survived and were retained for a longer period and were distributed within the degenerative IVDs. It is also important to understand that EVs released by hUC-MSCs transplanted into impaired discs may promote endogenous cell migration as well as the secretion of paracrine mediators that might promote IVD repair and regeneration [10,51]. Furthermore, it has been shown that NPCs and ECM components have a systematic regulatory effect on implanted hUC-MSC derived EVs, allowing cells to differentiate in vivo into a disc-like morphology [38]. These findings showed that the transplanted hUC-MSCs and EVs diffused, lived, and differentiated into the functioning NP lineages.

LBP is a common condition that arises at the initial phases of IVDD, as evidenced by X-ray and MRI analyses, and is hence associated with spinal cord injury; it also has a prominent inverse effect on DH. The major pathological indication of degenerative changes that radiographic images can detect is DH reduction [52,53]. The full penetrative needle puncture was radiographically evaluated using X-ray, which revealed significant IVD space reduction in the IVDD group compared to the control and cell transplanted groups. Moreover, preliminary findings of a rat IVDD model revealed that transplantation of hUC-MSCs and EVs after injury significantly increased the GAG content. This is consistent with the detection of GAG synthesis using the DMMB assay in previous degenerated IVD models [54]. Furthermore, changes in water content were investigated. The results revealed that the EV transplanted group was effective in restoring IVD hydration completely. The outcome was consistent with our findings, demonstrating that cell-free therapy successfully retrieved the water content [55]. As a result, treatments that increase GAG production or decrease GAG destruction are effective in restoring the water content and DHI of degenerated discs.

We investigated the effect of pain and inflammatory markers on degenerative and repair processes in NP that play critical roles in cellular survival, homeostasis, and chondrogenesis by analyzing genes to compare hUC-derived MSCs and hUC-MSC derived EVs transplanted groups. These involve *Adrenoceptor beta 2* (*ADRβ2*), *cartilage oligomeric matrix protein* (*COMP*), *cyclooxygenase-2* (*COX-2*), *preprotachykinin-A* (*PPT-A*), *C-X-C motif chemokine ligand 1* (*CXCL-1*), *matrix metalloproteinase 13* (*MMP-13*), and *chitinase-3-like protein 1* (*YKL-40*) markers. Our qPCR showed the mRNA expression patterns of these markers in healthy IVDs and degenerative discs transplanted with hUC-MSCs and EVs. We explored an overall profile that seems to be dependent on the period of IVDD. In the control IVDs, seven genes (*ADRβ2*, *COMP*, *CXCL-1*, *COX-2*, *PPT-A*, *MMP-13*, and *YKL-40*) were expressed. Elevated expression of *ADRβ2*, *CXCL-1*, *COX-2*, *PPT-A*, *MMP-13*, and *YKL-40* was observed during the establishment of a disc degenerative model with varied patterns at different degeneration intervals. In general, we observed a steady increase in the expression level as the time of deterioration increased. The expressions of *ADRβ-2* and *YKL-40* were reduced after 24 h compared to days 2 and 5, but increased after day 5 of degeneration. *COMP* expression was significantly increased following 24 h of degeneration. *COMP* expression level recovered to baseline after day 5. *COX-2* expression increased on day 1 and then steadily declined on days 2 and 5. *CXCL-1*, *PPT-A*, and *MMP-13* had high expression levels in the degenerative IVDs on days 1 and 2, but their levels steadily declined during the next 5 days of degeneration. It indicated that these genes mediate the degenerative IVD event in a time-dependent manner.

When hUC-MSCs and EVs were transplanted into degenerative discs, EVs transplanted group expressed those genes at reduced levels as compared to hUC-derived MSCs. It is mainly possible that the response of EVs to repair discs is due to a stronger immunosuppressive effect. On days 2 and 5, *ADRβ2* expression levels were reduced in both the EV and hUC-MSC treated groups, though EVs had lower *ADRβ2* expression than hUC-MSCs. The *COMP* pro-anabolic gene was significantly higher in both transplanting groups after day 2 and continued to rise until day 5 in both treatment groups. On days 2 and 5; however, *CXCL-1*, *COX-2*, and *PPT-A* levels in EV transplants were significantly lower than in hUC-MSCs. In addition, *MMP-13* expression was significantly reduced on day 2 following EV regeneration periods. Furthermore, by the end of the day 5 regeneration phase, *YKL-40* expression levels in EVs had returned to their basal level. Based on these findings, we can conclude that hUC-MSC and EV transplantation modulate the immune response and reduce pro-inflammatory mediator production in the acute inflammatory process of degenerative IVDs. As a result, these hUC-MSCs and hUC-MSC derived EVs were able to improve their survival capacity while maintaining the homeostasis of the disc. Nonetheless, the expression level of such genes varies as IVDD progresses.

MMPs and metalloproteinases with thrombospondin motifs (ADAMTSs) are the main proteases involved in matrix destruction in discs. Several MMPs and ADAMTS were shown to be elevated in human deteriorated discs [56]. Tumour necrosis factor alpha (TNF-α) and interleukin 1 beta (IL1-β) were identified as major pro-inflammatory cytokine mediators in the discs, because they promote the development of various pro-inflammatory mediators such as cytokines, chemokines, and MMPs [57]. CXCL-1 exhibits neutrophil chemotactic ability. It may have a role in the inflammatory process and have an autocrine effect on endothelial cells. A bioinformatic investigation revealed that the CXCL-1 gene may be involved in the pathogenesis of disc degeneration caused by an inflammatory response [58]. In all species, the ECM protein COMP was expressed at a lower level in NPCs than in AF cells. Though COMP has been found in the disc, its decreased expression may be due to variations in the mechanical characteristics of NP and CEP tissues [59]. COMP’s major molecular roles are to bind other ECM proteins and to catalyze the polymerization of collagen type II fibrils. Moreover, COMP has been shown to impede the vascularization of cartilage, which may also be the effect in disc tissues [60]. The interaction between COX-2 and YKL-40, which suggests the possibility of reciprocal impacts on IVDs, is intriguing. COX-2 is an enzyme that is involved in prostaglandin E_2_ (PGE2) formation and is thus associated with the pathology of lumbar disc degeneration [61]. Furthermore, COX-2 may have an indirect effect on chondrocyte metabolism since PGE2 suppresses aggrecan (ACAN) production [62]. Because YKL-40 functions similarly to a matrix-degrading enzyme, the actions of these two proteins may be synergistic, potentially accelerating the degenerative disc process. YKL-40 has been shown to have an important function in disc degeneration caused by local inflammation mediated by COX-2 and nitric oxide. YKL-40, a glycoprotein member of the “mammalian chitinase-like protein” family, has been extensively investigated for its potential role in IVD tissue remodelling [63]. It is a key secretory protein found in human chondrocytes. ADRβ2 signal increased with the level of IVD degeneration and is linked to changes in ECM expression. Research on chondrocytes and chondrogenic differentiated MSCs revealed that neurotransmitters, particularly norepinephrine, have a destructive effect on chondrocyte physiology and pathology, mostly through ADRα2a and ADRβ2 [64]. The *PPT-A* gene is considered to contribute to the progression of degenerative human disc disease by coding for substance P, supporting nociceptive sensitization, and modulating inflammatory reactions [65].

A key limitation of this study is the short evaluation period for two weeks post-transplantation. While this timeframe allows for assessment of early therapeutic responses, it does not capture the long-term regenerative potential, durability of benefits, or delayed adverse effects such as fibrosis or immune responses. Given that IVDD is a chronic and progressive condition, short-term improvements may not necessarily predict sustained therapeutic outcomes. This study also has some limitations that warrant consideration. First, EVs were derived from a single donor cell line, which, while ensuring consistency, does not account for donor-to-donor variability in EV composition and therapeutic potential. Second, the relatively small sample sizes in some experimental groups may reduce the statistical power of our analyses. Although the results reached significance, studies with larger cohorts are required to confirm the robustness and reproducibility of these findings. Degenerative IVD pathologies shows that the harsh IVD environment reduces the survival and lack of functional recovery of engrafted hUC-MSCs. To improve the treatment efficacy of hUC-MSCs and hUC-MSC derived EVs against impaired degenerative disc environments, the responses of pro-inflammatory mediators should be closely evaluated throughout degeneration. To the extent of our understanding, no previous research has specifically investigated the expression of pain and inflammatory markers after transplanting hUC-MSCs and EVs into degenerative IVD disease animal models. Hence, the outcome of NP renewal is based mainly on the collective outcomes of immunofluorescent, histologic, biochemical, and molecular analyses.

This study’s brief duration may therefore have limited the ability to adequately capture the intervention’s long-term effects or any potential delayed impacts on IVDD progression. According to earlier research, IVDD can have a clinical course that lasts for months or even years, during which time symptoms frequently change or get worse. To better understand the long-term effects of therapies and examine how they affect the long-term management of IVDD, future research should strive to include a longer follow-up period. This is especially crucial for determining if temporary gains are sustained and whether the illness worsens in spite of early treatment [44].

Notwithstanding the encouraging results of this investigation, it is critical to recognize the constraint imposed by the very brief follow-up duration. IVDD is a chronic, progressive illness that needs long-term therapeutic efficacy to be completely confirmed, even if our results show promising early regeneration responses after EV delivery. The stability of matrix restoration, long-term control of inflammatory responses, and the possibility of a delayed relapse of degeneration may not be sufficiently reflected by a brief monitoring period. Therefore, to thoroughly assess the long-term safety, integration, and regeneration impact of EV-based therapies within the complex disc microenvironment, future research should encompass longer follow-up periods, and maybe recurrent EV dosage will be crucial.

## 5. Conclusions

Our findings indicate that hUC-MSC-derived EVs contribute to structural restoration, attenuation of inflammation, and enhancement of matrix synthesis in degenerated intervertebral discs. These results underscore the therapeutic promise of EVs as a safe and effective cell-free strategy for treating IVDD. The two-week evaluation period limits our ability to assess the long-term efficacy of the intervention for IVDD. Given the chronic nature of IVDD, longer-term studies are needed to fully understand the sustained effects and disease progression. Moreover, detailed mechanistic studies are essential to validate these outcomes and facilitate clinical translation.

## Figures and Tables

**Figure 1 biomedicines-13-02420-f001:**
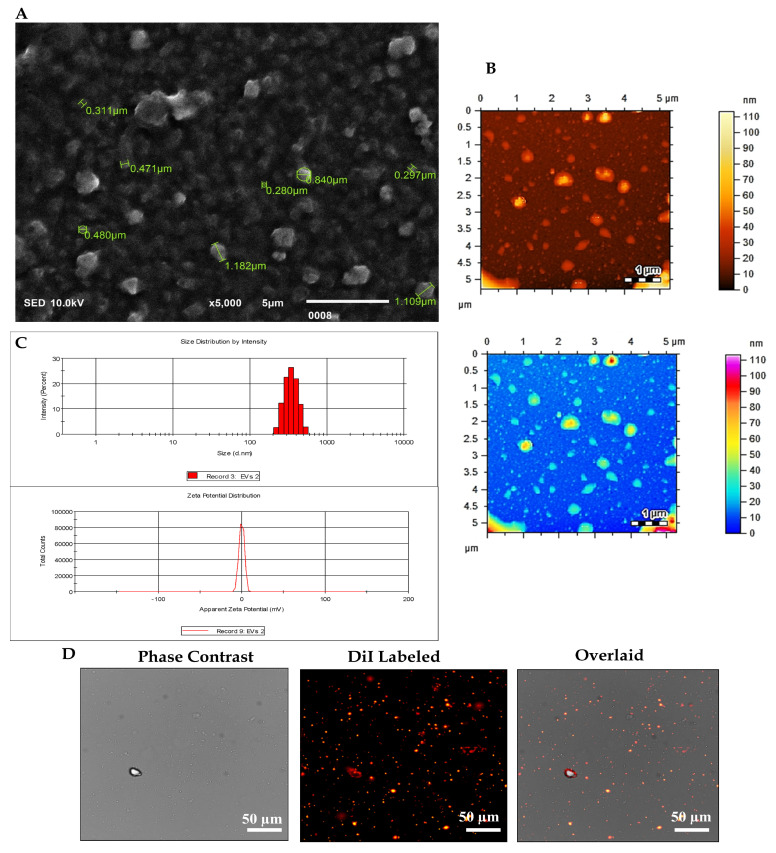
Characterization of EVs derived from hUC-MSCs. (**A**) SEM provides an image illustrating the geometry and size of the EVs. Scale bar = 5 μm. (**B**) AFM presents a topographic 2D image (amplitude modulation), demonstrating EV adhesion to mica, drying with N2, and captured in air. Scale bar = 1 μm. (**C**) DLS analysis using the Nano Zetasizer reveals the size distribution of EVs. The peak height indicates an average size of 403.3 ± 85.94 nm, while the zeta potential analysis shows a negative potential of approximately −0.270 ± 4.02 mV. A negative zeta potential suggests stability in an aqueous solution, with a value of 4.02 mV indicating stable aqueous dispersion in a physiological environment. (**D**) Light and fluorescent microscopy were utilized to evaluate the distribution of EVs isolated from hUC-MSCs Via the differential centrifugation technique. Scale bar = 50 μm. Overall, **A**–**D** provides a comprehensive overview of the physical characteristics, size distribution, surface properties, and molecular composition of the isolated EVs, enhancing our understanding of their potential biological functions and therapeutic applications.

**Figure 2 biomedicines-13-02420-f002:**
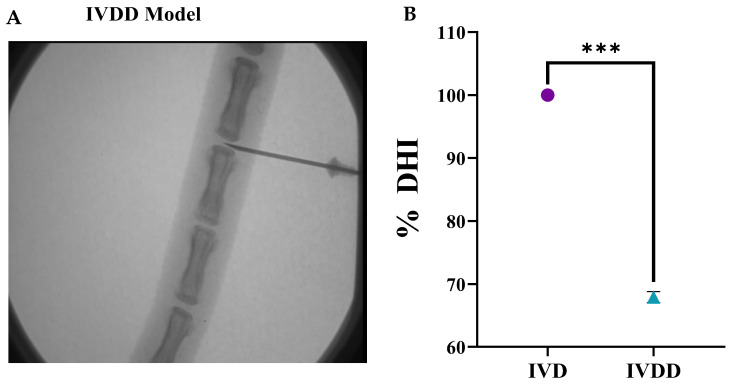
Development of In Vivo Rat IVD Degeneration Model. (**A**) Fluoroscopic imaging of the rat coccygeal spine showing puncture of Co [6/7] IVD using a 21G needle. (**B**) Disc Height Index (DHI) significantly decreased in punctured discs Co [6/7] compared to uninjured controls Co [5/6]. (**C**) qPCR analysis showing fold changes in inflammatory markers (*ADRβ2*, *COMP*, *CXCL-1*, *COX-2*, *PPT-A*, *MMP-13*, *YKL-40*) in IVDD vs. control discs across days 1, 2, and 5. Data are normalized to β-actin and shown as 2^−ΔΔCt^ fold change that statistically evaluated by one-way ANOVA and Bonferroni post hoc comparison. Significant increase in these genes was observed in contrast to untreated control as follows: Day 1: *COMP*, *PPT-A*, and *MMP-13* (** = *p* < 0.01) and *CXCL-1* and *COX-2* (*** = *p* < 0.001); day 2: *ADRβ2*, *CXCL-1*, *PPT-A*, and *MMP-13* (*** = *p* < 0.001); *COX-2* and *YKL-40* (** = *p* < 0.01); and *COMP* (* = *p* < 0.05); day 5: *ADRβ2*, *CXCL-1*, and *YKL-40* (*** = *p* < 0.001); *PPT-A* (** = *p* < 0.01); and *COX-2* (* = *p* < 0.05). The fold change is depicted on the *Y*-axis. The control IVD fold change is set to one. All results are presented as mean ± SD in triplicate, and the confidence level is *** = *p* < 0.001, ** = *p* < 0.01, * = *p* < 0.05, and ns = non-significant. *n* = 3 animals per group.

**Figure 3 biomedicines-13-02420-f003:**
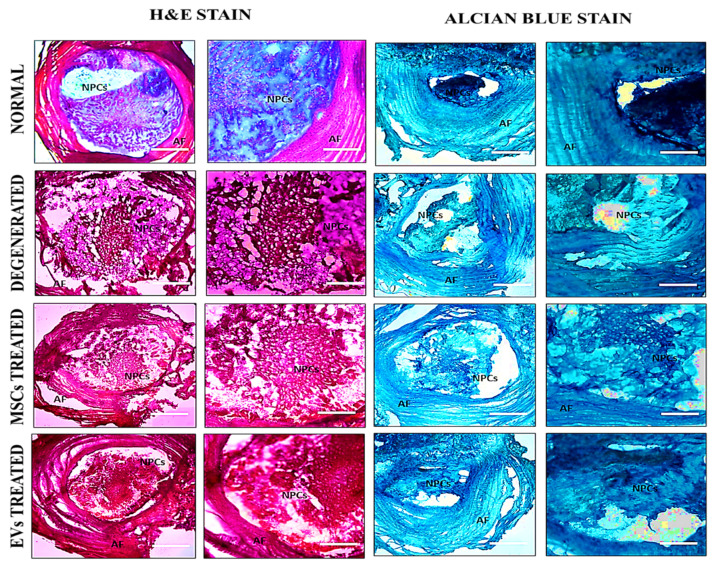
Histological Examination of Rat Healthy Normal, Degenerated, and Transplanted IVDs. Histological sections of Co5/6 (control), Co6/7 degenerated, and Co6/7 transplanted discs stained with Hematoxylin & Eosin (H&E) and Alcian Blue. Images captured at 10× (scale bar = 100 μm) and 20× (scale bar = 50 μm) magnifications. EV- and MSC-treated discs show partial restoration of nucleus pulposus (NP) structure and glycoprotein content compared to degenerated discs (NPCs, Nucleus pulposus cells, AF, Annulus fibrosus).

**Figure 4 biomedicines-13-02420-f004:**
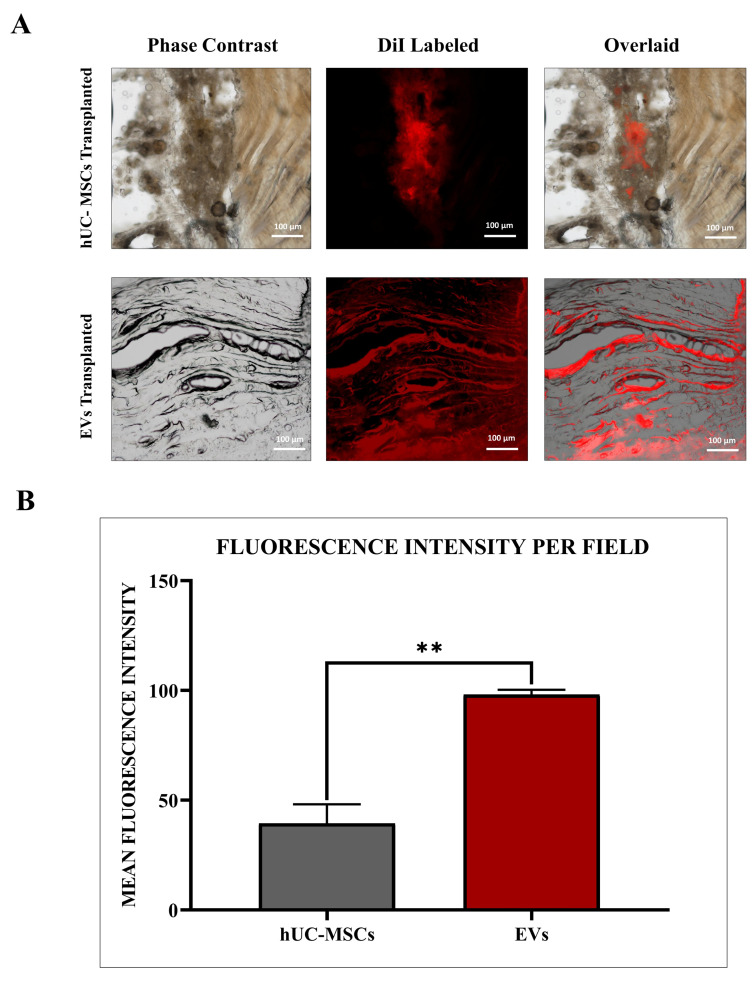
In Vivo Tracking of Transplanted Labelled hUC-MSCs and EVs. (**A**) Fluorescence microscopy showing DiI-labelled hUC-MSCs and EVs in punctured IVDs. (**B**) Quantification of mean fluorescence intensity per field in hUC-MSC and EV groups. Scale bar = 100 μm. All results are represented as mean ± SD in triplicate (*n* = 3 animals/group), and the confidence level is ** = *p* < 0.01.

**Figure 5 biomedicines-13-02420-f005:**
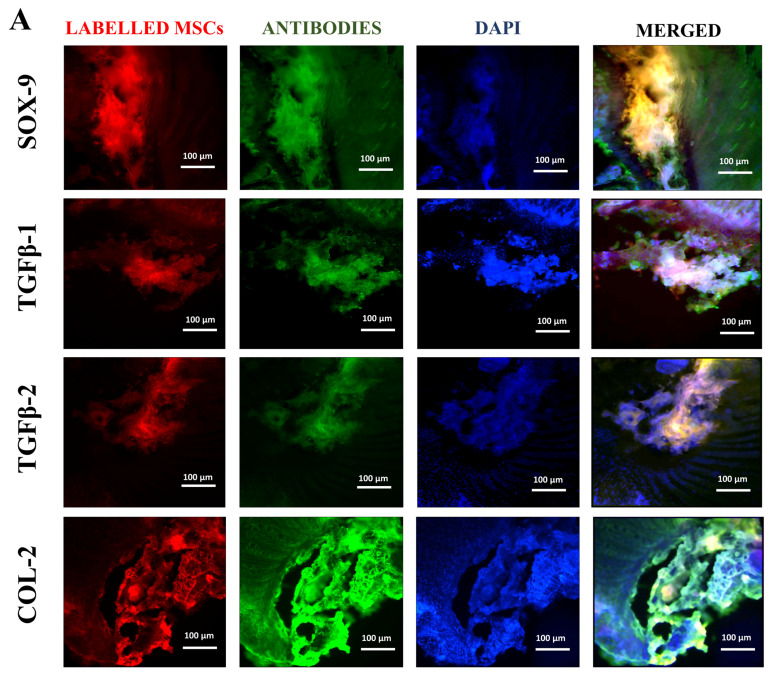
Immunohistochemically Evaluation of Labelled Human Chondrocyte Markers in Transplanted Discs. (**A**,**B**) Immunofluorescence images showing co-localization of DiI-labelled hUC-MSCs and EVs (red) with human-specific chondrocyte markers: SOX-9, TGFβ-1, TGFβ-2, and COL-2 (green). Nuclei stained with DAPI (blue). Immunohistochemical examination of the discs transplanted with EVs revealed significantly higher expression of human-specified chondrocyte markers, i.e., SOX-9, TGFβ-1, TGFβ-2, and COL-2, against labelled hUC-MSCs. Images captured at 10× magnification (scale bar = 100 μm). (**C**) Quantification of mean fluorescence intensity per field across groups. All results are represented as mean ± SD in triplicate (*n* = 3 animals/group), and the confidence level is *** = *p* < 0.001, and ** = *p* < 0.01.

**Figure 6 biomedicines-13-02420-f006:**
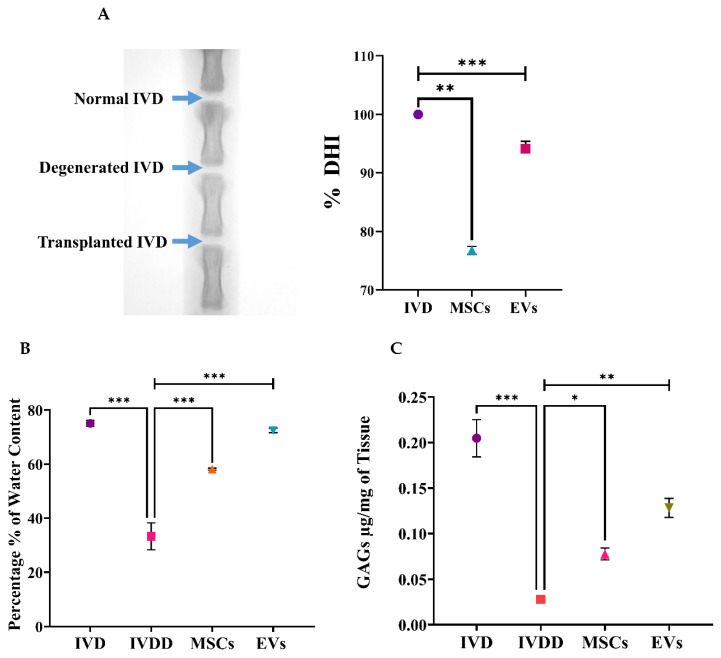
Functional Analysis In Vivo. (**A**) Fluoroscopic images of the rat coccygeal spine were taken with a 21G syringe needle used to puncture the Co [6/7] IVDs and transplanted the hUC-MSCs and EVs immediately after puncture. The % DHI value was increased in response to EV transplantation as compared to the hUC-MSC Co [6/7] IVDs. However, the % DHI value was significantly improved in EV-treated spinal discs as compared to hUC-MSC treated IVDs, followed by a significant decrease in the IVDD group. (**B**) The water content was determined using the heat evaporation approach. Water content in NP was measured in normal, degenerated, and treated discs with transplanted hUC-MSCs or EVs after 2 weeks of treatment. In comparison to the IVDD group, the disc transplanted with EVs showed significantly higher expression of water content in percentage. (**C**) GAGs content of the in vivo group was quantified by the DMMB assay. The IVD transplanted with hUC-MSCs and EVs showed a significantly upregulated expression in contrast to the IVDD group. For statistical analysis, the one-way ANOVA and a Bonferroni post hoc comparison were performed. All results are represented as mean ± SD in triplicate (*n* = 3 animals/group), and the confidence level is *p* ≤ 0.05 (*** = *p* < 0.001, ** = *p* < 0.01, * = *p* < 0.05).

**Figure 7 biomedicines-13-02420-f007:**
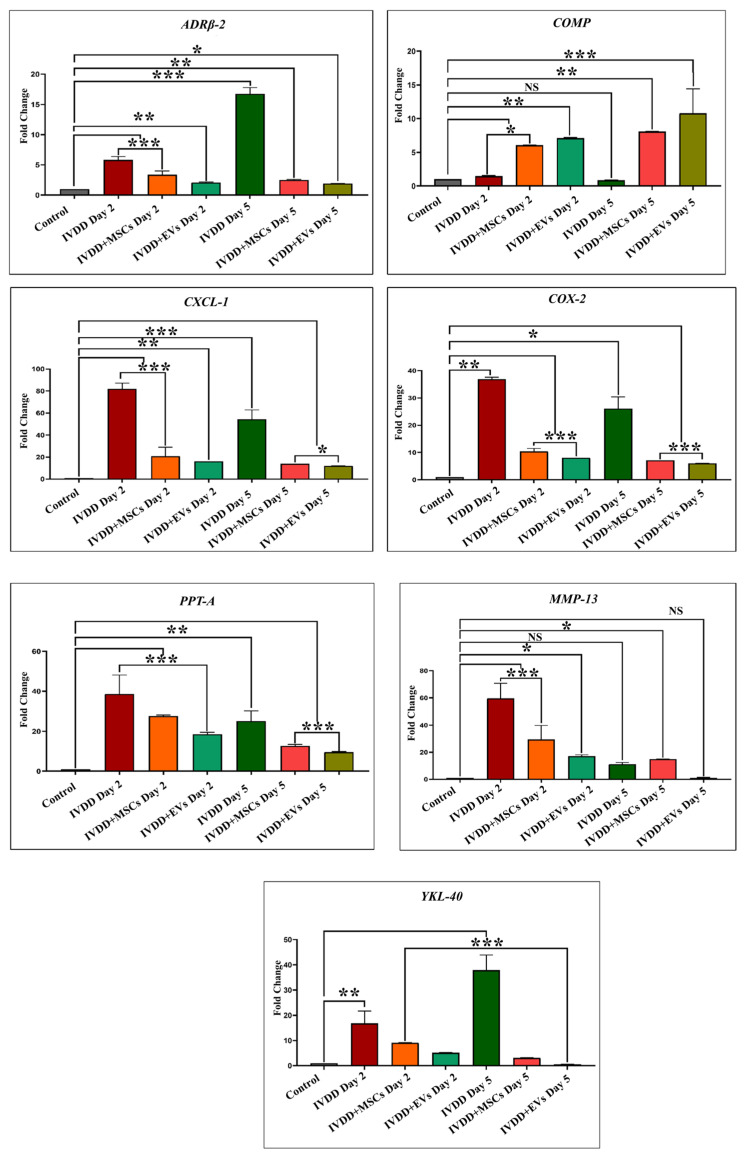
Gene Expression Analysis of Pain and Inflammation Markers in hUC-MSCs and EVs in an Implanted Disc Model. qPCR analysis of inflammatory markers *ADRβ2*, *COMP*, *CXCL-1*, *COX-2*, *PPT-A*, *MMP-13*, and *YKL-40* was performed in the control, IVDD, and transplanted groups. The bar charts indicate quantitative two-fold (2^−ΔΔCT^) changes in markers of inflammation statistically evaluated by one way ANOVA and Bonferroni post hoc comparison. The β-actin gene was used as a reference gene. For 2 and 5 days, hUC-MSCs and their derivatives, EVs, were implanted into the degenerated discs. *COX-2*, *PPT-A*, and *YKL-40* were significantly downregulated on days 2 and 5 (*** = *p* < 0.001) in hUC-MSC and EV treated groups. Some inflammatory genes, such as *ADRβ-2*, *CXCL-1*, and *MMP-13* showed significantly lower expression than degenerated IVDs: Day 2 (*** = *p* < 0.001); day 5: *ADRβ-2* (** = *p* < 0.01), and *CXCL-1* and *MMP-13* (* = *p* < 0.05) in hUC-MSCs in vivo; while days 2 and 5: *ADRβ-2* and *CXCL-1* (** = *p* < 0.01) and (* = *p* < 0.05); and day 2: *MMP-13* (* = *p* < 0.05) in EVs in vivo. However, only *COMP* revealed significant overexpression in both transplanted groups as compared to degenerated IVDs: Day 2 (MSCs * = *p* < 0.05, EVs ** = *p* < 0.01), day 5 (MSCs ** = *p* < 0.01, EVs *** = *p* < 0.001). The fold change is depicted on the Y-axis. The control IVD fold change is set to one. All results are presented as mean ± SD in triplicate, and the confidence level is *** = *p* < 0.001, ** = *p* < 0.01, * = *p* < 0.05, and NS = non-significant.

**Table 1 biomedicines-13-02420-t001:** List of primer sequences used in the study.

Gene Name	Gene ID	Primer Sequences F: Forward; R: Reverse	Product Size
*Adrenoceptor beta 2*	*NM_012492.2*	F 5′-ATTGCAGTGGATCGCTATGTT-3′ R 5′-GTGCATCTGGATAGGCAAGAA-3′	138
*Cartilage Oligomeric Matrix Protein*	*NM_012834.2*	F 5′-TCGGCTACATCAGGGTGC-3′ R 5′-TAGCGCAGGTTAGCCCAG-3′	142
*C-X-C Motif Chemokine Ligand 1*	*NM_030845.2*	F 5′-AGCACCATGGTCTCAGCC-3′ R 5′-ATCCCTGCCACTGTCTGC-3′	131
*Cyclooxygenase-2*	*AF233596.1*	F 5′-ATGACACAACAGCCCATCTCT-3′ R 5′-ATCCAGGCTGAACTCACACAT-3′	127
*Preprotachykinin-A*	*M14312.1*	F 5′-GGTGCCAACGATGATCT-3′ R 5′-GCATCCCGTTTGCCCATT-3′	152
*Matrix Metalloproteinase 13*	*NM_133530.1*	F 5′-ATTCTTCTGGCGTCTGCAC-3′ R 5′-GGGATGGATGCTCGTATGC-3′	110
*Chitinase-3-like protein 1*	*NM_053560.2*	F 5′-AAGAGCTTCACTCTGGCATCTT-3′ R 5′-ATATCTCGTAGTAGGCGAGGGT-3′	112
*Beta-actin*	*NM_031144.3*	F 5′-CAGCTCCTCCGTCGCC-3′ R 5′-CTCTGGGCCTCGTCGC-3′	220

## Data Availability

All the data are included in the manuscript.

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
