# Peer review of "Extracellular Vesicles Derived from Human Umbilical Cord-Mesenchymal Stem Cells Ameliorate Intervertebral Disc Degeneration"

_biomedicines, 2025, doi:10.3390/biomedicines13102420_

Round 1
Reviewer 1 Report
Comments and Suggestions for Authors
Dear authors,
Thank you for the opportunity to assess your manuscript.
The manuscript addresses a highly relevant topic in regenerative medicine: intervertebral disc degeneration. The authors explore the therapeutic potential of extracellular vesicles derived from hUC-MSCs. This is a clinically relevant and rapidly expanding area of research. The study is generally well designed, employs appropriate in vivo methodologies, and presents promising evidence supporting EVs as a cell-free therapeutic approach.
Strengths
- Novelty and significance: the comparison of hUC-MSCs versus their derived EVs for IVD repair is both timely and significant. The findings suggesting superior retention and reparative capacity of EVs add meaningful data to the field.
- Comprehensive methodology: the use of fluoroscopy-guided puncture, histology, qPCR, immunohistochemistry, and biomechanical assessments provides a robust dataset.
- Clear outcomes: the manuscript consistently shows that EVs outperform MSCs in retention, downregulation of inflammatory gene expression, and GAG/water restoration.
Having said that, I still think there are areas requiring some clarification and refinement before publication. Please find these comments immediately below.
Major comments
- Duration of study: the evaluation period of two weeks is relatively short. While the authors acknowledge this limitation, it should be emphasized more clearly in the discussion and conclusions, considering that IVDD is a chronic condition.
- Single donor source for EVs: EVs were obtained from a single donor line
This introduces bias and limits generalizability. Please comment explicitly on donor variability and whether future work will validate across multiple donors.
- Statistical power: some groups have very small sample sizes.
While technically acceptable, this weakens the robustness of the conclusions. The authors should acknowledge this limitation.
4. Mechanistic insights: the work shows phenotypic improvements, but the mechanistic claims such as immunomodulation via specific miRNAs or signaling pathways are speculative and not experimentally validated. Please soften the language in the discussion.
Minor comments
- Figures: some figure legends (Figures 2–5) are dense and could be simplified for readability. Ensure scale bars and statistical annotations are uniform.
- References: while generally appropriate, the manuscript could still benefit from the addition of more recent papers on EVs in IVDD.
- Formatting: ensure adherence to Biomedicines formatting guidelines.
Author Response
Response to Reviewer 1 is Attched

Reviewer 2 Report
Comments and Suggestions for Authors
This manuscript investigates the regenerative potential of extracellular vesicles (EVs) derived from human umbilical cord mesenchymal stem cells in a rat model of intervertebral disc degeneration (IVDD). The topic is of considerable interest in regenerative medicine, particularly as cell-free therapies are increasingly viewed as safer and more clinically feasible alternatives to direct stem cell transplantation. While the study provides valuable preliminary evidence, several important methodological and interpretive limitations reduce the strength of the conclusions. Addressing these concerns will substantially improve the translational relevance of the work. Here are few comments that need to be consider before publication.
Major Comments
- EV Characterization and Compliance with Standards
The study characterizes EVs using SEM, AFM, DLS, and DiI-labeling, which demonstrate morphology and size. However, no evidence is provided for EV-specific marker validation (e.g., CD9, CD63, CD81, Alix etc.), a critical requirement of the MISEV2018 guidelines (doi: 10.1080/20013078.2018.1535750). Without such validation, the preparation cannot be conclusively classified as EVs and may include a heterogeneous mixture of particles such as apoptotic bodies or protein aggregates. The authors are strongly encouraged to include at least a subset of canonical markers using Western blotting, nanoparticle tracking analysis (NTA), or flow cytometry, and to revise the methods section to explicitly note compliance with MISEV recommendations. - Source of EVs
The EVs were derived from a single donor cell line, which limits reproducibility and does not account for inter-donor variability. Such variability is important for translational relevance, as EV content and therapeutic activity can differ between donors. Please discuss this limitation in detail, and, if feasible, validate the findings using EVs derived from at least one additional donor. - Follow-Up Duration
The study endpoints extend only to two weeks post-transplantation. Given that IVDD is a chronic and progressive disease, this short timeframe reflects only acute responses. Long-term regenerative capacity, durability of benefits, and potential adverse effects (e.g., fibrosis, immune activation) remain unaddressed. An extended observation period would strengthen the study; at minimum, the discussion should explicitly acknowledge that short-term outcomes may not fully predict long-term therapeutic potential.
Minor Comments
- Terminology Consistency
The manuscript occasionally refers to “MSC-derived EVs” without specifying the umbilical cord origin. Please ensure consistent use of “hUC-MSC EVs” to avoid confusion with EVs derived from other MSC sources.
2.Results Overlap with Figure Legends
The Results section is currently presented in a figure-legend style, largely repeating what is shown in the images and graphs. It would benefit from a more integrative narrative that highlights major findings, compares treatment groups directly, and connects molecular, histological, and functional outcomes. This will improve readability and emphasize the study’s key advances rather than leaving the interpretation to the discussion.
Overall Recommendation
The manuscript presents promising preclinical findings; however, stronger EV characterization, acknowledgment of key limitations, and greater methodological rigor are required to meet publication standards.
Comments on the Quality of English Language
The sentences are generally complete and grammatical. However, several sections (particularly the Results and Methods) would benefit from language polishing.
Author Response
Response to Reviewer 2 is Attached

Round 2
Reviewer 2 Report
Comments and Suggestions for Authors
I thank the authors for addressing all the comments. The manuscript has improved considerably in clarity and presentation. I now find it suitable for publication.